# Plasmonic Metamaterial Ag Nanostructures on a Mirror for Colorimetric Sensing

**DOI:** 10.3390/nano13101650

**Published:** 2023-05-16

**Authors:** Sayako Maeda, Noboru Osaka, Rei Niguma, Tetsuya Matsuyama, Kenji Wada, Koichi Okamoto

**Affiliations:** Department of Physics and Electronics, Osaka Metropolitan University, Osaka 599-8531, Japannoboru9306@gmail.com (N.O.);

**Keywords:** plasmonics, metamaterials, localized surface plasmon resonance, colorimetric sensing, nanohemisphere-on-mirror

## Abstract

In this study, we demonstrate the localized surface plasmon resonance (LSPR) in the visible range by using nanostructures on mirrors. The nanohemisphere-on-mirror (NHoM) structure is based on random nanoparticles that were obtained by heat-treating silver thin films and does not require any top-down nanofabrication processes. We were able to successfully tune over a wide wavelength range and obtain full colors using the NHoM structures, which realized full coverage of the Commission Internationale de l’Eclairage (CIE) standard RGB (sRGB) color space. Additionally, we fabricated the periodic nanodisk-on-glass (NDoG) structure using electron beam lithography and compared it with the NHoM structure. Our analysis of dark-field microscopic images observed by a hyperspectral camera showed that the NHoM structure had less variation in the resonant wavelength by observation points compared with the periodic NDoG structure. In other words, the NHoM structure achieved a high color quality that is comparable to the periodic structure. Finally, we proposed colorimetric sensing as an application of the NHoM structure. We confirmed the significant improvement in performance of colorimetric sensing using the NHoM structure and succeeded in colorimetric sensing using protein drops. The ability to fabricate large areas in full color easily and inexpensively with our proposed structures makes them suitable for industrial applications, such as displays, holograms, biosensing, and security applications.

## 1. Introduction

Plasmonic color is one of the most intensively studied topics in plasmonic metamaterials [1]. Recently, full-color tuning of plasmonic colors has been expected to enable new plasmonic printing and display applications [2,3,4,5,6]. Many research groups have reported plasmonic colors using metallic nanostructures, such as nanodisk arrays [7,8,9,10], nanohole arrays [8,11,12,13], nano-antennas [14,15,16], nanorods [17,18], apertures [11], gratings [19,20,21,22,23,24,25,26,27,28,29,30], and metal–insulator–metal (MIM) resonators [31,32,33,34,35]. Full-color tuning using these various structures has been proposed. However, most of them are produced using top-down nanofabrication processes, which are costly and time-consuming and are not suitable for industrial applications.

Other methods, such as self-assembly [36], chemical synthesis [37], metal nanoparticle deposition [38], colloidal etching lithography [36,39], nanoimprint lithography [40], and laser interference lithography (LIL) [41], and nanosphere lithography [42], have been proposed for fabricating plasmonic metamaterials over large areas. However, there are some problems with these methods, such as inhomogeneity resulting in pastel colors. Furthermore, for display technology applications, the plasmonic metamaterials must be prepared over very large areas using simple and inexpensive fabrication processes. Ideally, the materials should also achieve full-color tuning and produce vivid colors.

Recently, we fabricated self-assembled 3D multilayer structures of Ag NPs synthesized by a bottom-up process and found that they exhibit unique optical properties when located on a metal substrate [43]. The peaks in the extinction spectra were split into two, enhanced, and made tunable due to the mode-splitting effect attributed to strong coupling. Quite recently, we also demonstrated the tuning of localized surface plasmon resonance (LSPR) in the visible range using nanohemisphere-on-mirror (NHoM) structures [44]. The NHoM structures have nanohemispheres arranged on a metal substrate with a dielectric spacer layer and do not require any top-down nanofabrication processes. The strong coupling between the SP modes in the metallic nanohemisphere and the mirror-image modes generated in the metallic substrate causes the SP resonance peak to split into two peaks. This phenomenon is based on the fact that this strong coupling induced electromagnetic effects. The detailed mechanism was explained in previous papers by comparing it with electromagnetic field analytical calculations [44]. We also succeeded in controlling the LSPR by using NHoM structures to tune the optical properties in the visible [44], deep UV [45,46], and near-IR [47] wavelength regions. In this paper, by simply changing the thickness of the dielectric spacer layer, we successfully tuned over a wide wavelength range and obtained full colors. The colors were analyzed on the chromaticity diagram of the XYZ color system and achieved full coverage of the Commission Internationale de l’Eclairage (CIE) [48] standard RGB (sRGB) color space.

Furthermore, by varying the dielectric spacer layer in steps, it is possible to create gradations of color. The ability to fabricate large areas in full color easily and inexpensively makes this structure suitable for industrial applications, such as displays, holograms, biosensors, and security applications. Moreover, we compared the random structure with a periodic structure. Although many plasmonic colors have been reported using periodic structures or structures without top-down nanofabrication processes, there have not been any reports comparing the two. We fabricated a periodic Ag nanodisk-on-glass (NDoG) structure using electron beam lithography and compared the spectra with the NHoM structure. The light scattering spectra of dark-field images observed by a hyperspectral camera at several observation points were measured to evaluate the variation of peak positions and shapes depending on the observation points. The results showed that the NHoM structure had less peak variation than the periodic Ag disk structure, despite using random hemispheres. In other words, the NHoM structures achieved a high color quality that was comparable to the periodic structures without using top-down nanofabrication processes.

Finally, we proposed colorimetric sensing as an application of the NHoM structure. We demonstrated that this structure enhances and sharpens the resonance spectrum, enabling high-sensitivity sensing with a large wavelength change range through calculations and experiments. Furthermore, the resonance peak wavelength of the silver NHoM structure is in the visible light region, allowing for the observation of refractive index changes as colors by the naked eye. We succeeded in highly sensitive colorimetric sensing of refractive index changes using this structure with protein drops. The results indicated that the colorimetric sensing based on the NHoM structure has potential for application as a highly sensitive biosensor.

## 2. Materials and Methods

### 2.1. Simulations

Finite difference time domain (FDTD) simulations were performed using commercial software (Poynting for Optics V03L 10R121, Fujitsu, Kanazawa, Japan). Periodic boundary conditions were set in the x- and y-directions, and absorbing boundary conditions were set in the z-direction. The model size for FDTD calculations using periodic structures was set to twice the diameter. Pulsed light consisting of a differential Gaussian function with a pulse width of 0.5 fs was used as the excitation light. The peak position on the excitation pulse spectrum was at approximately 600 THz (wavelength 500 nm). The dielectric function of Ag was approximated by Drude’s equation on the basis of the values reported by Johnson and Christy [43]. The refractive index of the glass was set to 1.5 to avoid dispersion. We used a non-uniform mesh with a grid size of 1–5 nm.

### 2.2. Sample Preparations

Samples of NHoM structures were prepared as follows: An Ag layer with a thickness of 100 nm was deposited on a cover glass by resistance thermal evaporation (SVC-700TM, Sanyu Electron, Tokyo, Japan), and an Al_2_O_3_ layer spacer was deposited by atomic layer deposition (ALD) (SAL1000, SUGA Co., Hokkaido, Japan) at various thicknesses (5–15 nm). The ALD was performed using water and trimethylaluminum (TMA) as precursors at a substrate temperature of 150 °C. The deposition rate was 1.14 nm/cycle. The thicknesses of Ag and Al_2_O_3_ were measured using a contact profilometer and an ellipsometer, respectively. A thin Ag layer was deposited on the spacer layer, and Ag nanoparticle (NP) structures were formed by heat treatment at 300 °C in an electric furnace in a nitrogen atmosphere. As a reference sample, Ag nanohemisphere-on-glass (NHoG) were also fabricated on a cover glass using the same method. When the thin Ag layer on top of Al_2_O_3_ was heated to 300 °C, this heating did not affect the other layers below, as confirmed by the fact that the reflection spectra did not change.

The sample of the NDoG structure, with a diameter of 100 nm and a height of 20 nm on a cover glass, was fabricated using an electron beam lithography system (ELS-7500EX, Elionix Inc., Tokyo, Japan). The resist of ZEP520A and the charge dissipating layer of Espacer 300Z were spin-coated and baked on the cover glass, and then the electron beam lithography was applied. After development in xylene for 3 min, 2 nm of Ni as an adhesive layer and 40 nm of Ag were deposited in order, and then lift-off was conducted using butanone.

### 2.3. Observations and Measurements

The surface morphology was investigated using atomic force microscopy (AFM) (NanoWizard, Bruker—JPK Instruments AG, Berlin, Germany) and high-resolution scanning electron microscopy (SEM) (FlexSEM 1000 II, Hitachi High-Tech, Tokyo, Japan). The transmission and reflection spectra were measured using a UV–visible spectrometer with a reflectance measurement attachment (5° incident angle, UV-2600, Shimadzu Kyoto, Japan) and were converted into extinction spectra. The light extinction (*E*) of the samples on the transparent glass substrates were converted as *E* = −log_10_ (*T*/*T*_0_), where *T* and T_0_ are the transmittances of the sample and substrate, respectively. On the other hand, *E* of samples on the metal substrate were converted as *E* = −log_10_ (*R*/*R*_0_), where *R* and *R*_0_ are the reflectance of the sample and substrate, respectively. The scattering spectra were measured using the dark-field images observed by a hyperspectral camera. We randomly obtained seven spectra for one pixel from the measured hyperspectral image (800 × 600 pixels) and compared the spectral differences depending on the observation points.

## 3. Results and Discussion

### 3.1. Design and Calculation for Vivid Color Tuning

Figure 1a,b show the sample structure and extinction spectra obtained from the FDTD simulation of Ag NHoM. The thickness of Ag metal layer was 100 nm, and the diameter of the nanohemisphere was set to 50 nm. The thickness and refractive index of the spacer layer were set to 15 nm and 1.5, respectively. The spectrum of the NHoM structure had peaks at 378 nm and 596 nm due to localized plasmon resonance. Figure 1c shows the spatial distribution of the localized electric field around the NHoM at the peak wavelengths (378 and 596 nm).

In Figure 1c, the short branch represents a quadrupole oscillation mode, and the long branch represents a dipole oscillation mode. Additionally, there were antisymmetric charges on the mirror substrate. The electrons on the metal film acted as if they canceled out the external electric field (mirror image charge). We consider that the two strong peaks in the NHoM structure are coupling modes of quadrupole and dipole oscillations with the mirror image mode of the Ag substrate, respectively. The antisymmetric charge coupling caused strong optical confinement, and the resonance peaks were stronger and sharper than in the NHoG structure.

In the NHoM structure, we found that the resonance peak is due to the interaction of the electric field between the hemisphere and the Ag substrate. This interaction depends on the distance between the hemisphere and the Ag substrate. The splitting energy of the SP mode strongly depends on the thickness of the spacer. As the spacer becomes thinner, the two modes move closer to each other, the mode coupling becomes stronger, and the splitting width increases. Conversely, as the spacer thickens and the distance between the two modes increases, the mode coupling disappears, and they return to their original single peak. Therefore, we attempted to control the resonance peaks by tuning the thickness of the dielectric spacer layer. Generally, changing the shape and size of the metal nanostructures is necessary to tune the surface plasmon resonance mode. However, if we can adjust it on the basis of only the thickness of the spacer layer without changing the shape and size of the metal nanostructure, it becomes very easy to tune the surface plasmon resonance flexibly over a large area.

Figure 2a shows the results of extinction spectra obtained from FDTD calculations by varying the thickness of the spacer from 5 nm to 50 nm for Ag hemispheres with diameters of 30 nm and 50 nm, respectively. The refractive index of the spacer layer was set to 1.5. Regardless of the diameter size, as the thickness of the spacer layer decreased, the distance between the nanohemisphere and the silver substrate became closer, and the two resonance peaks separated more. The longer wavelength side of the resonance peaks had a higher extinction, and the shorter wavelength side was mainly in the ultraviolet region. Therefore, the resonance peaks on the longer wavelength side significantly affected the sample’s color. For this reason, the longer branch was red-shifted as the spacer layer became thinner, and full-color tuning was achieved by adjusting the dielectric spacer layer.

To investigate the changes in chromaticity of the NHoM structure at Ag hemisphere diameters of 30, 50, and 160 nm, we analyzed the spectra calculated by FDTD and plotted the obtained colors on a chromaticity diagram of the XYZ color system. Figure 2b shows the transition in chromaticity coordinates as the Ag diameter of the NHoM structure changes. The chromaticity coordinates shifted near the origin as the diameter increased. As the diameter became larger, the spectral peak broadened, and the wavelength range of the spectra expanded, resulting in a pastel color near the center of the chromaticity coordinate due to color mixing. The dotted white area in Figure 2b represents the sRGB color space. Although a more pronounced color change appeared as the diameter decreased, the chromaticity at 30 nm in diameter existed outside the sRGB color space. By adjusting the size of the Ag NHoM structure, the chromaticity coordinate made a loop within the sRGB color space. This means that any natural color can be produced using this NHoM structure. As the thickness of the spacer layer decreased, the chromaticity coordinates rotated in a clockwise direction. This was due to the red shift of the resonance peak on the long wavelength side caused by the increase in the distance between resonance peaks, as shown in Figure 2a.

In the case of the 160 nm diameter, the short-wavelength peak was in the blue region, and the long-wavelength peak was in the infrared region. When the spacer layer was thinned, the resonance peak at short wavelengths shifted slightly toward the green region, the same as the peak at long wavelengths. To confirm the behavior of the two peaks when the thickness of the dielectric spacer layer was varied, we calculated the FDTD with a diameter of 50 nm and a thickness of the dielectric layer that varied from 5 to 50 nm. Figure 2c shows the relationship between the distance of the dielectric layer and the energy of the peak wavelength. As the thickness of the spacer layer increased, the resonance peaks on the short-wavelength side red-shifted, and on the long-wavelength side, they blue-shifted. As the thickness of the dielectric layer increased, the two peaks became closer and converged to a single resonance peak, as in the spectra of the Ag NHoG structure. The thicker dielectric layer separated the hemispheres from the silver substrate and eliminated the interaction of localized charges, so that the spectrum of the NHoM structure with a thicker dielectric layer was equivalent to that of the Ag NHoG structure.

Figure 2d shows the color palette of the spectra obtained for the NHoM structure with diameters ranging from 30 to 90 nm. The refractive index of the spacer layer was set to 1.5. The color palette also shows a red shift in color with increasing Ag hemisphere diameter and decreasing dielectric layer thickness. The blue coloring at diameters of 90 nm and dielectric thicknesses of 10 nm or less resulted from the shift of the peak on the long-wavelength side to the infrared region. When the diameter of the Ag hemisphere was further increased, the reflectance spectrum became broad, and it was difficult to distinguish a clear difference in color.

Thus, we found that the NHoM structure causes the LSPR resonance to be trapped between the Ag nanohemispheres and the Ag layer, resulting in a strong interaction between them. The major difference between the NHoM structure and the MIM (metal–insulator–metal) structures, reported as plasmonic metamaterials in recent years, can be seen in the extremely thin spacer layer that confines the light strongly in a very small space.

### 3.2. Experimental Verification of Color Tuning using Random Structures

Next, we fabricated NHoM structures and attempted to obtain the three primary plasmonic colors of red, green, and blue. The obtained colors were compared with the results of FDTD calculations. We analyzed reflectance spectra obtained from FDTD simulations using LabSolutions (Shimadzu, Kyoto, Japan) for chromaticity analysis. The absorption, transmission, and reflection were defined as R(λ) + A(λ) + T(λ) = 1, where R(λ) is the spectral reflectance, A(λ) is the spectral absorption, and T(λ) is the spectral transmittance. In the Ag NHoM structure, T(λ) was set to 0 because the specular substrate was too thick to transmit visible light. The plasmonic color is produced by absorption and scattering of light due to the LSPR. Therefore, the chromaticity of the NHoM structure was analyzed using the spectral absorption A(λ) = 1 − R(λ).

First, we produced the sample with the LSPR resonance wavelength in the red region by setting the thickness of Al_2_O_3_ and Ag thin film to 7.5 nm and 7 nm, respectively. Figure 3a shows an AFM image of the red sample surface. Figure 3b shows a comparison of the extinction spectrum of the red sample and the spectrum obtained by FDTD calculation. The resonance spectrum showed good agreement between the experimental extinction spectrum and the FDTD results. Therefore, as shown in Figure 3b, the calculated color from the FDTD simulation and the dark-field image were also in good agreement. The experimental peak intensity on the long-wavelength side was weaker and broader than that in the calculation. The AFM image in Figure 3a showed that the grain size of the Ag nanohemispherical structure was random. Since the NHoM structure was fabricated by heat treatment without using top-down nanofabrication processes, the sizes of the particles were random, resulting in the broadening of the spectrum.

Next, we produced a sample with the LSPR resonance wavelength in the green region by setting the thickness of Al_2_O_3_ and Ag thin film to 15 nm and 7 nm, respectively. Figure 3c shows a comparison of the extinction spectrum of the green sample and the spectrum obtained by FDTD calculation. The resonance spectrum showed good agreement between the experimental extinction spectrum and the FDTD results. As shown in Figure 3c, the calculated color from the simulation and the dark-field image were close.

Finally, we produced a sample with the LSPR resonance wavelength in the blue region by setting the thickness of Al_2_O_3_ and Ag thin film to 5 nm and 16 nm, respectively. Figure 3d shows a comparison of the extinction spectrum of the blue sample and the spectrum obtained by FDTD calculation. In this experiment, we deposited a 16 nm Ag thin film, more than twice as thick as in the previous experiment. The size of the Ag nanohemispheres became extremely large, and the resonance peak on the short-wavelength side, which was in the ultraviolet region, moved to the visible region. The mode coupling was strengthened by making the dielectric layer thinner, and the resonance peak on the long-wavelength side was shifted to the infrared region, while the peak on the short-wavelength side was shifted to the visible region. As a result, we succeeded in obtaining a peak in the blue region and realizing the blue color.

From these results, we realized that RGB colors can be produced using the NHoM structure, and we succeeded in fabricating the three primary colors, which means that full color can be experimentally obtained by combining the hemisphere size and the thickness of the spacer layer.

We compared the FDTD results for tuning of LSPR with the experimental results. The thickness of Al_2_O_3_ was varied from 5 to 15 nm in 2.5 nm increments. The silver hemisphere was set to 35 nm in diameter for the FDTD calculations, while in the experiments, the Ag thin film was set to 7 nm and annealed. Figure 4a shows the extinction spectra calculated by FDTD and the calculated colors, and Figure 4b shows the extinction spectra and microscope images obtained experimentally. We measured the microscopic images in bright field for the NHoG structure and in dark field for the NHoM structure. In both cases, scattered light was measured, so the colors of the spectral peaks were observed in the microscopic images. The simulated and experimental spectra showed good agreement. The calculated colors and the microscopic images were also in good agreement. From these results, we have demonstrated both experimentally and numerically that it is possible to tune color over a wide wavelength range by changing only the thickness of the dielectric spacer layer in the NHoM structure. Furthermore, by varying the dielectric spacer layer in steps, it is also possible to create gradations of color.

### 3.3. Comparison of Random Structures and Periodic Aligned Structures

In addition, we compared the random structure with a periodic structure. We actually fabricated a periodic Ag NDoG structure using electron beam lithography and compared the spectra with the NHoM structure. Figure 5 shows the SEM image of the NHoG structure (a), the NHoM structure (b), and the NDoG structure (c) with a diameter of 100 nm and height of 20 nm. Figure 5d–f show the scattering spectra measured at any seven locations in the dark-field images of the NHoG, NHoM, and NDoG structures. The histogram of the diameter of the nanohemispheres of the NHoM structure shown in Figure 5d and of the nanodisks of the NDoG structure shown in Figure 5e are shown and compared in Appendix A.

In the NHoG structure, the peak wavelength varied greatly by observation points, while that of the NHoM structure varied little. The peak of the NHoM structure is due to the coupling mode of dipole oscillation with the Ag substrate, which reduced the change in the peak position due to the hemisphere size. The NHoM structure is random, but the variation in peak position was very small, approaching that of the NDoG structure.

We fitted the spectra of all pixels from hyperspectral images (800 × 600 pixels) of the NHoG, NHoM, and NDoG structures. Figure 5g–i show graphs of peak wavelength versus intensity. We evaluated the variation of resonance wavelengths at high peak intensities for each structure. Figure 5g shows that the peak wavelength variation was significant for the NHoG. In contrast, the peak wavelength variation was small for NHoM in Figure 5h. Figure 5i shows that the NDoG structure fabricated by electron beam lithography also had a small variation in peak wavelength.

In the NHoM structure, the peak wavelength variation was suppressed by the coupling between the silver hemisphere and the bulk metal layer. Therefore, the NHoM structure achieved vivid plasmonic color and full-color tuning without using any nanofabrication process.

### 3.4. Colorimetric Sensing and its Potential for Biosensor Applications

We proposed the NHoM structure as a colorimetric sensing and demonstrated its ability to enhance and sharpen the resonance spectrum, as well as achieve high-sensitivity sensing with a large wavelength change range through both calculations and experiments. We first analyzed the electromagnetic field using the FDTD method. Figure 6a shows the model and extinction spectra we calculated for a refractive index sensing with a sample on the NHoM structures. The extinction spectra were calculated for hemispherical diameters of 50 nm and peripheral refractive indices of 1, 1.5, and 2. The resonance peak wavelengths were red-shifted as the refractive index around the hemisphere increased, due to changes in the charge balance between the bulk metal layer and the hemisphere caused by changes in the refractive index around the hemisphere. Furthermore, these spectra were significantly affected by the thickness of the spacer layer in the NHoM structures. Thus, we calculated the dependence of the spectra on the thickness of the spacer layer. Figure 6b shows the sensitivity and full width at half maximum (FWHM) for various thicknesses of the spacer. The sensitivity was calculated by dividing the shift in peak wavelength by the change in refractive index. The FWHM of the spectra varied greatly with the thickness of the spacer layer, but the sensitivity of this structure increased linearly and did not change significantly with the thickness of the spacer layer. From these results, we calculated the sensing performance FOM by dividing the sensitivity by the FWHM. The maximum values of the FOM were 12.7 when the optimal spacer layer was used near the peripheral refractive index of 1, which was higher than other structures using the LSPR. This confirms the significant improvement in sensing performance by optimization.

Next, we fabricated NHoM structures with a thickness of Al_2_O_3_ and Ag thin film set to 15 nm and 7 nm, respectively, and annealed them. For comparison, we prepared NHoG structures with an Ag thin film set to 7 nm and annealed. On top of these, we deposited 20 nm of SiO_2_ using a high-vacuum RF sputter to mimic biological materials, which typically have a refractive index of 1.5, and measured the spectral change due to the refractive index change. Figure 6c and d show the experimentally obtained extinction spectra of the fabricated NHoM and NHoG structures. In the NHoM structure, the peaks at 368 nm and 531.5 nm are resonant peaks due to the coupling modes of quadrupole and dipole vibrations with the mirror image mode of the Ag substrate, respectively. These peaks were sharper than those obtained for the Ag NHoG. When the peripheral refractive index was changed from 1 to 1.5 by depositing SiO_2_ on the NHoM structures, the peak wavelengths were red-shifted to 389.5 nm and 577 nm, respectively. This peak shift was also larger than that of the Ag NHoG, and more sensitive sensing could be achieved. In addition, the SiO_2_ deposition shifted the quadrupole peak from the UV wavelength range to the visible range, and the dipole peak also broadened, resulting in a significant change in the sample surface color from pink to blue. This demonstrates that the NHoM structures can be used as highly sensitive colorimetric sensing.

Finally, we demonstrated colorimetric sensing using protein drops. Figure 6e shows the experimental results of the colored sensing applications. We added a 5 mg/mL drop of human serum albumin (HSA) to the NHoM and NHoG structures and glass slide. HSA drops caused the peripheral refractive index to change from 1 to 1.5, and the color of the sample surface changed significantly from pink to blue in the NHoM structures sample. In the NHoG structures, there was no change in color due to the drop of HSA. These results show that the NHoM structure can be applied to highly sensitive colorimetric biosensing.

Based on the findings, the NHoM structure is a promising candidate for use as a colorimetric biosensing due to its ability to achieve highly sensitive sensing over a wide range of wavelengths. Unlike the periodic structure, the NHoM structure is random but still displays a small variation in peak position. The traditional method of using lithography on various Ag disks systematically to obtain RGB colors is a lengthy and expensive process, but with the NHoM structure, colors can be adjusted by changing the thickness of the spacer in a random structure without the use of lithography.

Furthermore, it was recently reported that lithographically fabricated NDoG structures can also be color-tuned systematically and flexibly by applying heat treatment [49]. Although nanodisks on a mirror substrate may have even more interesting optical properties, they are difficult to fabricate and are currently being attempted and will be reported in the near future. Overall, these findings offer valuable insights into the potential applications of NHoM and NDoG structures in the field of colorimetric biosensing.

Although our NHoM structure shows promise for biosensing, it is important to detect a variety of actual samples, calculate the limit of detection (LOD) and limit of quantification (LOQ), and determine the dynamic range of sample concentration in order to demonstrate its potential for sensor applications. These practical efforts will be reported in detail in the near future.

## 4. Conclusions

In this paper, we reported that full-color tuning of the LSPR in the visible region can be achieved using the NHoM structure. This structure is based on random nanoparticle structures obtained by heat-treating silver thin films and does not require any top-down nanofabrication processes. Chromaticity analysis showed that all colors in the sRGB color space can be created using the NHoM structure, and the RGB three primary colors were successfully obtained in experiments. Furthermore, color tuning over a wide range of wavelengths can be achieved by simply changing the thickness of the dielectric spacer layer. Therefore, we have shown that plasmonic color can be easily and inexpensively fabricated over a large area by using the NHoM structure. Additionally, we fabricated a periodic structure using electron beam lithography and compared it with the random-based NHoM structure. The analysis of hyperspectral images showed that the NHoM structure had less variation in the resonant wavelength by observation points compared with the periodic NDoG structure. In other words, the NHoM structure achieved as high color quality as the periodic structure, even though it did not use any top-down nanofabrication processes. Finally, we proposed colorimetric sensing as an application of the NHoM structure. We confirmed the significant improvement in sensing performance using the NHoM structure and succeeded in colorimetric sensing using protein drops. The ability to fabricate large areas in full color easily and inexpensively is suitable for industrial applications, and the structure discussed in this paper can be applied to displays, holograms, biosensors, and security applications.

## Figures and Tables

**Figure 1 nanomaterials-13-01650-f001:**
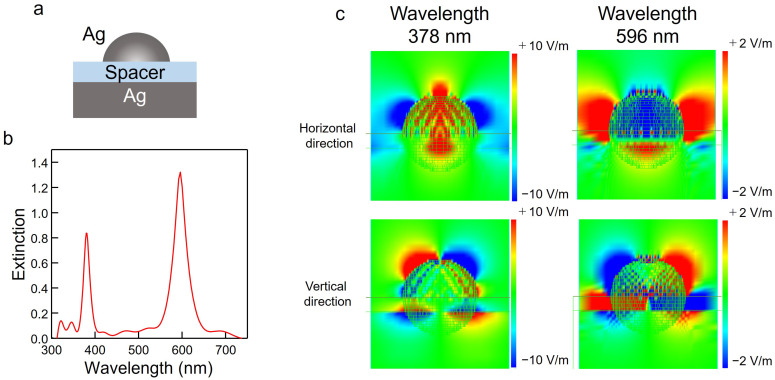
Extinction spectra and electric field spatial distribution of the nanohemisphere-on-mirror (NHoM) structure calculated using the FDTD method. (**a**) Schematic of the NHoM structure. (**b**) Calculated extinction spectrum of the NHoM structure. (**c**) Spatial distribution of the electric field around the NHoM structure at 378 nm and 596 nm.

**Figure 2 nanomaterials-13-01650-f002:**
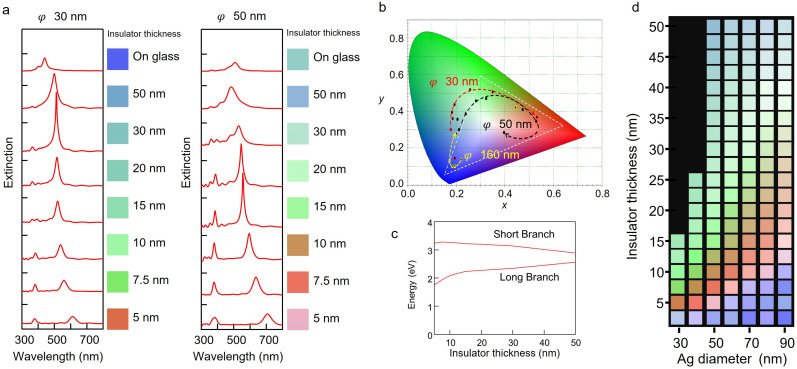
(**a**) Dependency on the extinction spectra and reflection colors as a function of spacer thickness (from 5 to 50 nm) of the NHoM structure by FDTD calculations. (**b**) Transition in chromaticity coordinates of reflection colors as the Ag diameter of the NHoM structure changes. (**c**) Relationship between peak wavelength of short and long branches and Al_2_O_3_ spacer layer thickness from 5 to 50 nm. (**d**) Color palette of the spectra obtained for the NHoM structure with diameters from 30 to 90 nm.

**Figure 3 nanomaterials-13-01650-f003:**
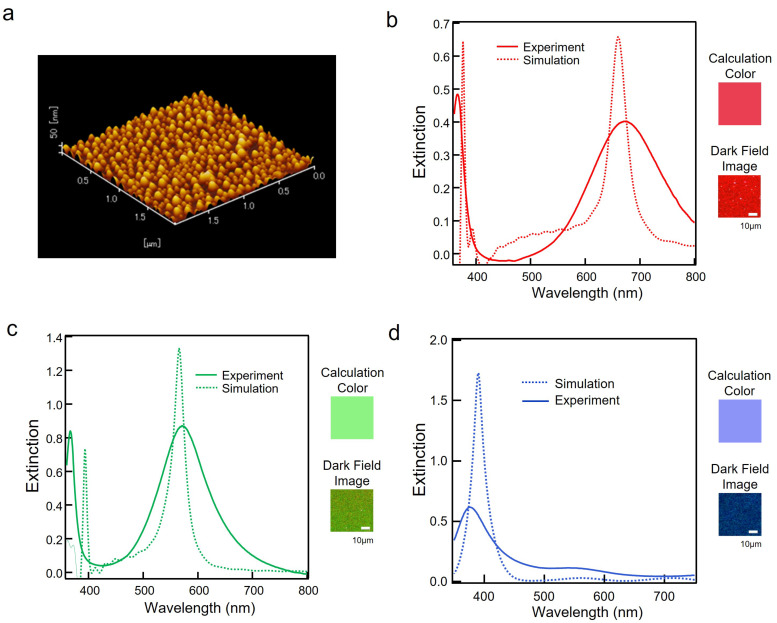
(**a**) A 3D-AFM image of the NHoM structure fabricated with the LSPR resonance wavelength in the red region, with the thickness of Al_2_O_3_ and Ag thin film set to 7.5 nm and 7 nm, respectively. (**b**) Comparison of the experimental and calculated extinction spectra of the NHoM structure with the LSPR resonance wavelength in the red region. (**c**) Comparison of the experimental and calculated extinction spectra of the NHoM structure with the LSPR resonance wavelength in the green region, with the thickness of Al_2_O_3_ and Ag thin film set to 15 nm and 7 nm, respectively. (**d**) Comparison of the experimental and calculated extinction spectra of the NHoM structure with the LSPR resonance wavelength in the blue region, with the thickness of Al_2_O_3_ and Ag thin film set to 5 nm and 16 nm, respectively.

**Figure 4 nanomaterials-13-01650-f004:**
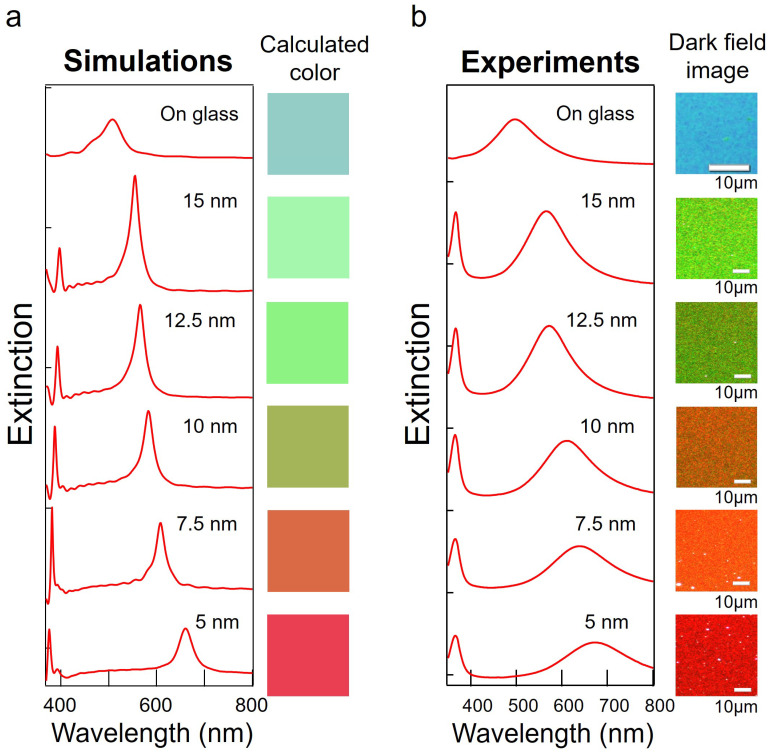
(**a**) The dependency of the extinction spectra and reflection colors on the spacer thickness (ranging from 5 to 15 nm) of the NHoM structure, as calculated by FDTD. (**b**) The experimental and calculated results of the extinction spectra and dark field images as a function of spacer thickness (ranging from 5 to 15 nm) of the NHoM structure.

**Figure 5 nanomaterials-13-01650-f005:**
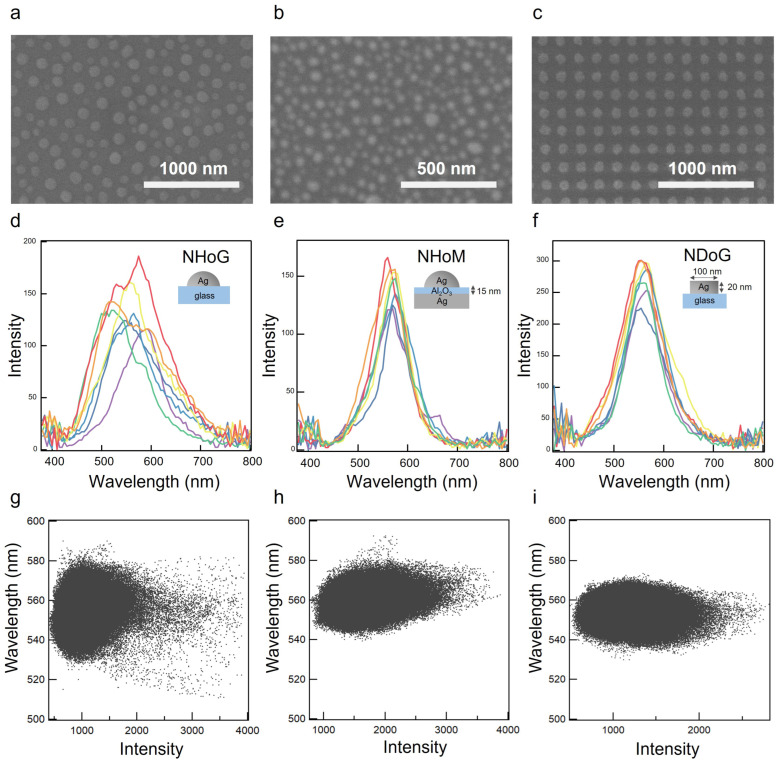
Top view of the SEM image of the nanohemisphere-on-glass (NHoG) structure (**a**), the NHoM structure (**b**), and the nano-disk-on-glass (NDoG) structure (**c**). Dark-field scattering spectra at any seven locations of the NHoG (**d**), NHoM (**e**), and NDoG (**f**) structures using a hyperspectral camera. Fitting results for peak wavelengths and intensities of spectra obtained from hyperspectral images (800 × 600 pixels) of the NHoG (**g**), NHoM (**h**), and NDoG (**i**) structures.

**Figure 6 nanomaterials-13-01650-f006:**
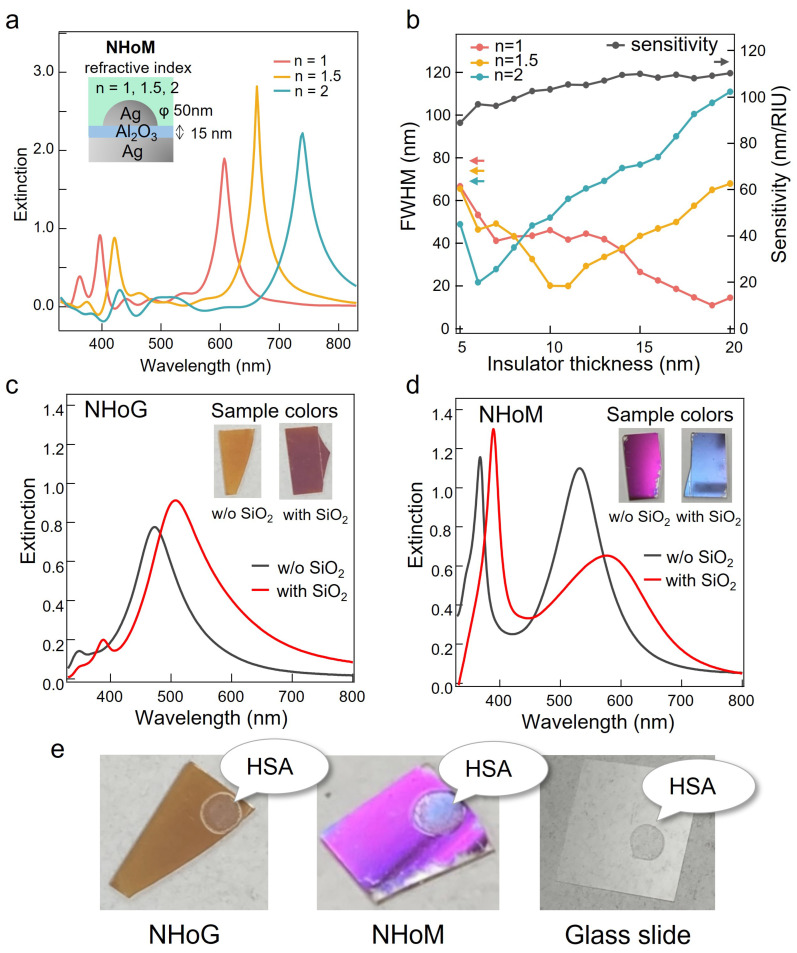
(**a**) Model and extinction spectra calculated for hemispherical diameters of 50 nm and peripheral refractive indices of 1, 1.5, and 2 using FDTD calculations for a refractive index sensing on NHoM structures. (**b**) Sensitivity and full width at half maximum (FWHM) for various thicknesses of the spacer. Experimentally obtained extinction spectra and sample colors of fabricated NHoG (**c**) and NHoM (**d**) structures. (**e**) Sample colors of NHoG and NHoM structures and glass slide with a 5 mg/mL drop of human serum albumin (HSA) as a refractive index change.

## Data Availability

The data presented in this study are available on request from the corresponding author.

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
