# Peer review of "Plasmonic Metamaterial Ag Nanostructures on a Mirror for Colorimetric Sensing"

_nanomaterials, 2023, doi:10.3390/nano13101650_

Round 1

Reviewer 1 Report

The publication entitled, ‘Plasmonic Metamaterial Ag Nanostructures on a Mirror for 2 Colorimetric Sensors’, by Maeda et al., describes a method of creating a colorimetric sensor based on Ag discs on a dielectric spacer. The work presented has interest. I however have many concerns with the publication and I have put my views below. Firstly, it was very difficult to read the manuscript and refer to the figures that were at the end of the manuscript. The figures should appear immediately after their first mention in the manuscript.

1-      What are the growth conditions for atomic layer deposition of Al2O3?

2-      How did the authors estimate the thickness of the various layers: Ag, Al2O3, Ag? Please provide SEM image in cross section.

3-      If the thin Ag layer on top of Al2O3 was heated to 300C, what effect did this heating have on the other layers below?

4-      Please provide the formula to convert transmission spectra into extinction spectra.

5-      Please explain the mechanism associated with the change in chromaticity and spacer thickness. Is it quantum tunnelling? The surface plasmon is produced at the contact of Ag with Al2O3, how about the volume plasmon in big Ag nanoparticles of 50 nm? Also, we need to understand the merging of the 2 extinction peaks for thick spacer layer.

6-      Very clearly, along with the thickness of Al2O3 layer, it is also important to control the size of the Ag disks. The sharpness in color is dependent on the uniform size distribution of the disks. The authors should discuss this and suggest means to produce standard disks. Please keep in mind that lithography is expensive.

7-      Please provide size distribution histograms of the Ag disks obtained for each sample

8-      Why was SiO2 deposited by sputtering? The refractive index on both sides of Ag disks should be the same in order to have mirror properties. Why was Al2O3 not chosen as the top layer?

9-      Why was lithography not used for the various Ag disks systematically to obtain RGB colors?

Reviewer 2 Report

The presented work summarizes a new approach for the design of a novel 2D plasmonic sensor. The work is interesting but I have some comments and questions.

1) The authors are experts in the preparation of 2D nanomaterial systems. Nonetheless, they do not refer to their previous work in any references. Although I support the suppression of the excessive self-citations in the scientific articles, I think looking at the results of 1-2 previous own articles in the Introduction (e.g. doi: 10.35848/1882-0786/ac2632, doi: 10.35848/1882-0786/abee63) would have been appropriate in order to verify the continuity of the presented research work.

2) I recommend that the Figures should be presented near the relevant sentences. The continuous scrolling in the manuscript to match the text and the figure is very confusing. 

3) Only one SEM image was presented in the manuscript about the NDoG structure (on Fig. 5b). The scale bar cannot be seen due to the small size of picture. I recommend that the SEM and AFM pictures from both NHoM and NDoG structures should be presented with larger-scale bars in the main text.

4) Though, the authors presented one possible application of the developed sensor glasses. The described results are not satisfactory. I have some comments to promote potential sensor uses.

          a) First of all, besides the HSA protein, several interference molecules should be also tested (other proteins, DNA fragments...). It is necessary to show the selectivity of the 2D optical sensor.

          b) I recommend testing the possible sensor applications in the case of real samples (e.g. blood serums or artificial blood). It is necessary to show the real usability.

          c) It is necessary to calculate the limit of detection (LOD) or the limit of quantification (LOQ) in the case of the HSA molecule (or other detectable analytes).

          d) It is important to determine the dynamic range of the HSA concentration.

In summary, the presented work is good but I saw some deficiencies. I suggest a major revision before the publication.

Round 2

Reviewer 1 Report

The paper can be accepted it this form.

Author Response

Thank you very much for accepting our paper in this form.

I deeply appreciate your kind peer review and helpful suggestions.

Reviewer 2 Report

Most recommendations are done but unfortunately, the sensor application is still not suitable. In the most cases, the measurements do not need a lot of time for the calculation of the limit of detection. Without the determination of LOD or LOQ values the name "sensor" cannot be used! I understand the future perspectives to use the presented platform in real samples but I do not believe in the separation of these works. Without these measurements, I cannot recommend the publication in present form. 

I suggest to ask more time from the Editor to present the works in only one manuscript!

Round 3

Reviewer 2 Report

The suggested modification have been corrected in light of the wordings/terms. Since the work is novel and may be of great interest, I support publication in light of the corrections.